# Obstacle Detection and Safely Navigate the Autonomous Vehicle from Unexpected Obstacles on the Driving Lane

**DOI:** 10.3390/s20174719

**Published:** 2020-08-21

**Authors:** Malik Haris, Jin Hou

**Affiliations:** School of Information Science and Technology, Southwest Jiaotong University Xipu campus, West Section, High-tech Zone, Chengdu 611756, Sichuan, China; malikharis@my.swjtu.edu.cn

**Keywords:** roadway hazard, Markov random field, autonomous vehicle, deep learning, image processing, self-driving car

## Abstract

Nowadays, autonomous vehicle is an active research area, especially after the emergence of machine vision tasks with deep learning. In such a visual navigation system for autonomous vehicle, the controller captures images and predicts information so that the autonomous vehicle can safely navigate. In this paper, we first introduced small and medium-sized obstacles that were intentionally or unintentionally left on the road, which can pose hazards for both autonomous and human driving situations. Then, we discuss Markov random field (MRF) model by fusing three potentials (gradient potential, curvature prior potential, and depth variance potential) to segment the obstacles and non-obstacles into the hazardous environment. Since the segment of obstacles is done by MRF model, we can predict the information to safely navigate the autonomous vehicle form hazardous environment on the roadway by DNN model. We found that our proposed method can segment the obstacles accuracy from the blended background road and improve the navigation skills of the autonomous vehicle.

## 1. Introduction

Global Status Report on Road Safety is released by World Health Organization (WHO, Geneva 27, Switzerland) 2018, in which WHO claims that about 1.35 million people die each year in road traffic accidents [1,2]. Similarly, American Automobile Association (AAA) Foundation released press report in 2016 that 50,658 vehicle roads accidents occurred only in America from the year 2011 to 2014 due to roadway obstacles. Roadway obstacles were the main factor of vehicle crashes and caused 9850 injuries and 125 deaths annually in the United States [3]. Reports indicate that over 90% of crashes are caused by errors of driver [4]. To improve this situation, governments, municipal departments, and car manufacture companies have considered significant investments to support the development of various technologies such as autonomous vehicles and cognitive robots. About 1 billion euros already have been invested by EU agencies on such type of projects [5].

In 2009, autonomous vehicles were developed and tested in four different states in the United States under the supervision of companies such as Waymo (Google, Mountain View, CA, U.S) and Uber with the support of traditional car manufacturers such as Ford and BMW [6]. Since then, this technology has been evolved and currently it is introduced in 33 different states of the United States with its specific regulations. In addition, Victoria Transport Policy Institute quoted that this technology will be widely used after 2040–2050 [7]. Currently, the most advanced features found in autonomous vehicles are Lane Changing (LC) Control [8,9], Adaptive Cruise Control (ACC) [10], Automatic Emergency Braking System (AEBS) [11], Light Detection and Ranging (LIDAR) [12], Street Sign Recognition [13], Vehicle to Vehicle (V2V) Communication [14], Object or Collision Avoidance System (CAS) [15], etc.

With the continuous development of the highways, roads, city streets, and expressways, challenging problems are increasing to distinguish because the road environment is complex and constantly developing. It will be influenced by small obstacles or debris, shadows, light, water spots, and other factors. Those objects have been fallen from vehicles, construction sites or are littering. Different types of sensors, active sensors (RADAR or LIDAR) to passive sensors (camera), were used to solve this problem. Active sensors such as RADAR or LIDAR offer high precision in measuring distance and speed from point to point but they often suffer from low resolution and high costs. However, in comparison of passive sensors such as camera, its accuracy is crucial for the timely detection of small obstacles and an appropriate response by safety critical moving platforms. Detecting the small obstacle that is displayed in a very small area of the image with all possible shapes and forms is also very challenging problem. Gradient induced by the edges of obstacles can also be caused by paper or soil dust on the road or due to moisture gradient after a rain, mud, or road marking, which can be potential sources of false positives. Figure 2 describes the phenomena very well.

In recent research and development, Convolutional Neural Network (CNN) models are used in autonomous vehicles to navigate safely. For example, during training, the CNN-based end to end driving model maps a relationship between the driving behavior of humans using roadway images collected from stereo camera and the steering wheel angle [16,17,18], and during testing, the CNN-models predict the steering wheel angle to navigate the autonomous vehicle safely. So that the autonomous vehicle is depended on the training dataset. If the CNN model is not trained on the roadway obstacle than navigation system of autonomous vehicle may generate incorrect information about the steering wheel angle and cause a collision in result. In addition, the number of research studies shows that the autonomous vehicle navigation system may fail to navigate safely due to several reasons, such as Radar sensor failure, camera sensor failure, and software failure [19]. For example, Tesla Model 3 failing to stop for an overturned truck and slamming right into it on highway in Taiwan even ignores the pedestrian with autopilot on.

This study addresses how to improve the robustness of obstacle detection method in a complex environment, by integrating a Markov random field (MRF) for obstacles detection, road segmentation, and CNN model to navigate safely [20]. We segment out the obstacle from the image in the framework of MRF by fuses intensity gradient, curvature cues, and variance in disparity. After analyzing the obstacle from the captured image, CNN-model helps to navigate the autonomous vehicle safely. The main contributions of this study are as follows:Pixel label optimization of images as a small obstacle or hindrance on the road detected by using an MRF model.Navigating an autonomous vehicle on a roadway from unexpected obstacle.

The remaining part of the research is organized as follows:Section 2—Reviews the relevant works carried out and developed in the past few years.Section 3—Introduces the method for detecting the physical obstacles or hindrances on the road and predicts the steering wheel angle for AV.Section 4—Shows demonstration and simulation.Section 5—Discusses the results and its comparison.

## 2. Related Work

The probability of occupation map is one of the main directions of work for obstacle detection [21]. It is developed through orthogonal projection of the 3D world onto a plane road surface (assuming that the environment structured of the road surface is almost planar). The plane is discretized in cells to form a grid; therefore, the algorithm predicts the probability of occupation of each cell. We conclude that this method can accurately detect the large obstacle (such as cyclists or cars) by using the stereo vision sensor [22] and can also help to identify the road boundaries by using LIDAR data [23]. However, since the probability function is closely related to the number of measurements in one grid, this method may not be suitable for small obstacle detection by using the stereo vision if the observation is scarce and noisy.

Digital Elevation Map (DEM) [24] is one of the algorithms that tries to detect obstacles relying on the fact that they protrude up from a dominant ground surface. The obstacle detection algorithm proposed in [24] marks DEM cells as road or obstacles, using the density of 3D points as a criterion. It also involves fitting a surface model to the road surface. Oniga and Nedevschi [25] presented a random sample consensus (RANSAC)-based algorithm for road surface estimation and density base classifier for obstacle detection by using the DEM constructed on stereo data, whereas in [26], authors used a similar RANSAC-based methodology for curb detection by using the polynomial fitting in stereo DEM. Although, RANSAC-based algorithm is not suitable for detection of small obstacle because the small obstacle and variation of disparity in the image is often similar to the noise levels and a least square fitting may smooth out the small obstacle along with road surface, RANSAC shows an accurate estimate of the vehicles position relative to the road [27].

Fusion of 3D-LIDAR and high-resolution camera seems to be a reasonable approach for robust curb detection. Traditional range visual fusion methods [28,29] use the detection results from the range data to guide the curb searching in the image, which have the advantage of enlarging the detection spot. However, fusion of 3D-LIDAR and visual data search small obstacle in the visual images is more reliable in enhance the robustness of obstacle and edge detection as compared to traditional range visual fusion methods [30]. It recovers a dense depth image of the scene by propagating depth information from sparse range points to the whole high-resolution visual image [31]. J. Tan and J. Li et al. [32] used the following geometric properties to robustly curb detection such as depth image recovery, curb point detection, curb point linking and curb refinement, and parametrization.

Scene flow-based approach is also used for obstacle detection [33,34], where each point in constructed 3D cloud is tracked temporally and the flow is analyzed to classify the object in the image such as road surface, obstacle, pedestrian, etc. This approach has limited applicability because it just detected the moving obstacles such as moving vehicle, bicycle, or pedestrians. In addition, the decision-making process is based on the flow of nearby points in 3D cloud, which can be too sparse for small obstacles. For obstacle detection in [35], authors used advanced stereo-based algorithms, combining full 3D data with motion-based approach, and yet it focuses only on large obstacle detection such as vehicles.

Hadsell et al. [36] presented the method of vision-based neural network classifier. His idea was to use an online classifier, which is optimized for long-distance prediction and deep stereo analysis to predict the obstacles and to navigate vehicles safely. Ess A. et al. [37] defined the method of combining image-based category detectors with its geometric information received from stereo cameras (such as vehicle or pedestrian detection). However, due to the large differences in its shape, size, and visual look, it will become difficult for vision sensor to train the dataset on small obstacles along the way.

Bernini and Bertozzi et al. [38] proposed an overview of several stereo-based generic obstacle detection such as Stixels algorithm [39] and geometric point clusters [40,41]. The Stixels algorithm distinguishes between a global ground surface model and a set of obstacle segments in rectangular vertical, thus, providing a compact and robust representation of the 3D scene. However, geometric relation between 3D point is used to detect and cluster obstacle point. This method is suitable for detecting medium-sized obstacle over close or medium distance. If the distance increases and the size of the obstacle decreases, then position accuracy and obstacle detection become challenging.

Zhou J. and J. Li [42] proposed a solution to detect obstacles using a ground plane estimation algorithm based on homography. This work extends [43] to a smaller obstacle by combining several indicators, such as homography score, super-pixel segmentation, and line segment detector in an MRF framework. However, such indexation based on appearance such as line segment detection and pixel segmentation fail when they occurred. Therefore, we directly use the lowest level of details available, i.e., image gradients in both stereo depth images and appearance.

Many researchers [44,45,46] work on the hypothesis of a flat surface assumption, free space, or the ground as a single planar and characterize the obstacles according to their height from the ground. The geometric deviation from the reference plan can be estimated directly from the image data, precalculated point cloud [47], or by extracting from v-disparity histogram model [48]. Instead of relying on the flatness of the road [49], the vertical road profile is modeled as a clothoid [50] and splines [51], which is estimated from the lateral projection of the 3D points. Similarly, free ground profile model was examined using adaptive threshold values in v-disparity domain and multiple filter steps [52].

In addition, we also explore existing datasets that are used in autonomous vehicle community. The dataset is provided by the Udacity (Mountain View, CA, USA) [53], which supports the end-to-end data and image segmentation, but it does not provide the ground truth for the obstacle in the roadway lane. Similarly, KTTI [54] and Cityscape [55] datasets also do not support the obstacle detection as a ground truth data. The dataset which matches our requirements is the Lost and Found dataset [56].

Pinggera P. and Kanzow C. have worked on planar hypothesis testing (PHT) [57,58], fast direct planar hypothesis testing (FPHT), point compatibility (PC) [40,41], Stixels [39,59], and mid-level representation: Cluster-Stixels (CStix) [60,61,62] for detecting the small obstacle in dataset. In addition, Ramos and Gehrig et al. [63] further extends this work by merging the deep learning with Lost and Found dataset hypothesis results. The people who have worked on Lost and Found dataset [56] did not discuss that how to navigate an autonomous vehicle safely or predict the steering wheel angle for an autonomous vehicle from unexpected obstacle on a roadway.

Since the existing dataset is not able to meet our requirements, we have carefully created our dataset using the steering angle sensor (SAS) and ZED stereo device mounted on an electric vehicle as discuss in the experimental setup section. After that, pixel labelling of small obstacles or hindrance on the road are detected by using MRF model and road segmentation model, which help the CNN-model to navigate the autonomous vehicle on roadways from unexpected obstacle by prediction the steering wheel angle.

## 3. Method

In this section, we will describe how to develop a safe autonomous driving system in unexpected and dangerous environments. In this section, we discussed three models.
(a)In the first model, we used various stochastic techniques (such as curvature prepotential, gradient potential, and depth variance potential) to segment obstacles in the image from Markov random field (MRF) frames. These three techniques measure pixel-level images to extract useful information and store it for orientation. In this method, each pixel in the node of interest was distributed in the MRF. Finally, instead of using OR gates, we used AND gates to combine the results of previous techniques.(b)In the second model, semantic segmentation technology was used to segment paths and filter outliers and other important obstacles.(c)Third model was used to predict the steering wheel angle of the autonomous vehicle. We analyzed the unexpected obstacle on the roadway and determined the threat factor ($T_{\mathrm{tf}}$). This threat factor helped us to ignore that obstacle or consider as accident risk.

The overall pipeline is graphically illustrated in Figure 1. The system input consisted of optical encoders used in applications for angle detection as a SAS in vehicles and two stereo images, which were used to calculate depth using the SGBM algorithm (semi-global block matching) proposed by Hirschmuller [64]. The depth variance and color images were used to calculate three different cues, i.e., image gradient, disparity variance, and depth curvature. These three cues were combined to a unary potential in an MRF with an additional pairwise potential. Then, the standard graph cuts were used to obtain the obstacle. In addition, deep learning-based semantic segmentation [65] was used to segment the roads and filter out the abnormal values.

### 3.1. The Markov Random Field (MRF) Model

During the past decade, various MRF models, inference, and learning methods have been developed to solve many low-, medium-, and high-level vision problems [66]. While inference concern underlying the image and scene structure to solve problem such as image restoration, reconstruction, segmentation, edge detection, 3D vision, and the object labeling in the image [67]. In our research work, we elaborate our query regarding small obstacle segmentation from the road by defining an energy function (such as cost) over an MRF. We associate the image with random process *X* with the elements $X_{s}$, where s ϵ S represents a position of a pixel in the image. Each pixel position in the image is considered as a node in the MRF and affiliates each node with the unary or pairwise cost [68]. Minimum energy function E is defined in equation as:(1)$$E\left(X\right)=\;\underset{s}{\sum }E_{u}\left(s\right)+\;\underset{\left(r,s\right)\epsilon N}{\sum }E_{p}\left(X_{r},\;X_{s}\right)$$
where $E_{u}\left(s\right)$ represents the unary term and $E_{p}\left(X_{r},\;X_{s}\right)$ represents the pairwise term. $X=\left\{X_{1},\;X_{2},\;\ldots ,\;X_{n}\right\}$ is set of random variables associated with the set of nodes S that takes label $X_{s}$ ϵ [0, 1], which depends on the nature of appearance, either it is road, texture, or the small obstacle on the road. Around each pixel, we calculate the unary term $\left(E_{u}\left(s\right)\right)$ independently. It is the combination of three cues such as gradient potential $\left(E_{u}^{g}\left(s\right)\right)$, depth curvature potential $\left(E_{u}^{c}\left(s\right)\right)$, and depth variance potential $\left(E_{u}^{v}\left(s\right)\right)$. The unary term $\left(E_{u}\left(s\right)\right)$ is defined as follows:(2)$$E_{u}\left(s\right)=E_{u}^{g}\left(s\right).E_{u}^{c}\left(s\right).E_{u}^{v}\left(s\right)$$

### 3.2. Gradient Potential

The potential gradient at the site (*i*, *j*) is defined as follows:(3)$$E_{u}^{g}\left(i,j\right)=\sqrt[{2}]{G_{x}{\left(i,\;j\right)}^{2}+G_{y}{\left(i,\;j\right)}^{2}}$$

Partial derivative ($G_{x}\left(i,j\right)$ and $\;G_{y}\left(i,j\right)$) is calculated in the horizontal and vertical direction in the original color image [69]. In our work, we use the image gradients rather than edge detectors, because edge detectors mostly perform the thresholding on the feeble gradient response, whereas our work avoid such types of thresholding, and it builds several weak cues as a strong cue.

### 3.3. Curvature Prior Potential

Curvature can be defined as the reciprocal of the radius of a circle that is tangent to the given curve at a point. Curvature is “1” for a curve and “0” for a straight line. Smaller circles bend more sharply and hence have higher curvature. This feature is very useful for autonomous vehicle in multiple purposes such as free space detection, curb detection, etc. [70,71]. When performing an Eigen analysis near the 3D point under consideration, the algorithm greatly quantifies the changes along the normal surface. Equation of the curvature potential is defined:(4)$$E_{u}^{c}\left(s\right)=\frac{\sqrt[{2}]{\lambda_{3}}}{\sqrt[{2}]{\lambda_{1}}+\sqrt[{2}]{\lambda_{2}}+\sqrt[{2}]{\lambda_{3}}}$$
where $E_{u}^{c}\left(s\right)$ is a curvature prior at the corresponding position obtained after reprojection the 3D point in the image space. The $\sqrt[{2}]{\;}$ of the eigenvalue ($\lambda_{1},\lambda_{2},\lambda_{3})$ can better distinguish nonplanar surfaces. This technique is little different than the M. Pauly and M. Gross et al. [70] algorithm. The curvature prior feature is stable and robust and stable than tangent plane normal vectors because it does not make a plane surface assumption very clearly.

### 3.4. Depth Variance Potential

If we slide horizontally on the image, we can calculate this potential to detect sudden changes in depth [72]. This is computed by summation of these sudden depth horizontal windows ($W_{h}$) in a square window ($W_{s}$) and multiplying the figured-out value with corresponding disparity in the pixel, even the obstacles are not available. Equation of depth variance is defined as:(5)$$D_{var}\left(i,j\right)=var\left(\left[D\left(i-\frac{W_{h}}{2},j\right)\;:\;D\left(i+\frac{W_{h}}{2},j\right)\right]\right)$$

Here, the depth map is represented by *D*, which we obtained with the help of stereo equipment. The final variance potential is defined as follows:(6)$$E_{u}^{v}\left(i,j\right)=\;\left(\overset{W_{s}/2}{\underset{a=\;-W_{s}/2}{\sum }}\overset{W_{s}}{\underset{b=0}{\sum }}D_{var}\left(i+a,j-b\right)\right)\;.\;d\left(i,j\right)$$

Here, parallax value of the specific located pixel (*i*, *j*) is represented by d(i,j). 

The results of above three potentials are shown in Figure 2. Thus, we can see that the curvature prior potential and depth variance potential are reliable. However, adding some noisy depth values to the stereo will result in false positive values, as shown in Figure 2b,c. To get the more accurate unary potential result, as seen in Figure 2e, weighting the gradient potential in Figure 2d can pacify the problem by using the depth potential.

### 3.5. Pairwise Potential

By using the Potts model, we define the pairwise potential $E_{p}\left(X_{r},X_{s}\right)$:(7)$$E_{p}\left(X_{r},X_{s}\right)=-J_{p}\underset{\left(r,s\right)}{\sum }\delta \left(X_{r},X_{s}\right)$$

In the above equation, Kronecker delta is represented by $\delta \left(X_{r},X_{s}\right)$; whenever $X_{r}=X_{s}$, the value is 1 otherwise 0. The equation of finding the global minima of energy function of the small obstacle that is detected on the road is:(8)$$X^{*}=\begin{array}{c} argmin\;E\left(X\right)\\ x \end{array}$$

With the help of graph cut feature, we can find the global minima efficiently, whereas minimum graph cut is found using Boykov Y. and Kolmogorov V. et al. [73] procedure of min-cut/max-flow. Appendix A provides the pseudocode for obstacles detection by using the MRF-model.

### 3.6. Determination of the Obstacle Threat Value in the Image

In this model, we determine the threat value ($T_{tf}$) by detecting obstacles in the capture image. Coordinate (*x, y*) of the obstacle in the capture image is identified once we combine the result of semantic segmentation model and MRK mode. We can compute the threat value ($T_{tf}$) by measuring the distance of the obstacles in the image size (*height* (*h*) and *width* (*w*)) from the bottom center pixel ($h,\frac{w}{2}$). Finally, the threat value is computed from the following formula:(9)$$T_{tf}=1-\;\sqrt{\frac{{\left(x-h\right)}^{2}+{\left(y-\frac{w}{2}\right)}^{2}}{h^{2}+{\left(\frac{w}{2}\right)}^{2}}}$$

To obtain the latitudinal and longitudinal distance of the obstacle from the center line, we subtract the height (*h*) with half of the width ($\frac{w}{2}$) values from x and y values, respectively. Code related detection of obstacle in the captured image whose pixels size >= 50 along with it coordinate specified in Appendix A.

### 3.7. DNN-based Autonomous Vehicle Driving Model

In our research, we have implemented an autonomous driving model similar to DAVE-2, i.e., an end-to-end autonomous driving model [18]. Here, we train the weights of the network to minimize the average square error between the steering command issue. As shown in Figure 3, the network receives an obstacle detection and labeling image of 400 × 600 × 3 pixels and generates a steering wheel angle as output. This network architecture consists of 11 layers, including 1 lambda layer, 1 normalization layer, 5 convolution layers, and 4 fully connected layers. We use 5 × 5 kernel and 2 × 2 stride in the first three convolution layers and 3 × 3 kernel and 1 × 1 stride used in last two convolution layers. The whole network contains 7,970,619 trainable parameters.

We train autonomous vehicle models based on the result of obstacle detection by MRF and SAS and then test their performance through prediction of steering wheel angle as describe in Algorithm 1. After training, our well-trained autonomous vehicle model detects high threat value obstacle on the roadway and navigate the autonomous vehicle safely. Appendix A shows the model generation code used for steering wheel angle prediction.
**Algorithm 1** Pseudocode for Predicting the Steering Angle#**Nvidia Model**
**Lambda**: Output shape: 400 × 600 × 3Image **normalization** to avoid saturation and make gradients work better.#**2D Convolution Neural Network for handle features.**
**Convolution1**: *5 × 5, filter: 24, strides: 2 × 2, activation: ELU***Convolution2**: *5 × 5, filter: 24, strides: 2 × 2, activation: ELU***Convolution3**: *5 × 5, filter: 48, strides: 2 × 2, activation: ELU***Convolution4**: *3 × 3, filter: 64, strides: 1 × 1, activation: ELU***Convolution5**: *3 × 3, filter: 64, strides: 1 × 1, activation: ELU*#**Dropout avoids overfitting**
*Drop out (0.5)*#**Fully Connected Layer for predicting the steering angle.**
**Fully connected 1**: *neurons: 100, activation: ELU***Fully connected 2**: *neurons: 50, activation: ELU***Fully connected 3**: *neurons: 10, activation: ELU***Fully connected 4**: *neurons: 1 (output)**model.compile(Adam(lr = 0.0001), loss=’mse’)**return model*

## 4. Dataset

Since little attention has been paid to finding small obstacles in the past, no reasonable dataset is available for public use. Therefore, we created a new small dataset for obstacle detection. The dataset contains 5626 stereo images. These images are obtained from ***ZED stereo device***, which were installed on electric vehicles in various environments. ZED camera can create the real-time point cloud at a framerate of 30 fps with the resolution of 2 × (1280 × 720), and we extracted them from different video sequences of electric vehicle view, while running on road. Specification of the ZED camera is presented in Table 1.

***Optical encoders*** or separate arrays of LEDs and photodiodes are used as steering angle sensor (SAS) in vehicle angle sensing applications [74,75]. Accurate and reliable steering angle detection is the main challenge of modern vehicles. This data is used not only for electronic power steering (EPS) and electronic stability control (ECS) but also for controlling adaptive headlights, lane keeping assist system, or other advanced driver assistance systems (ADAS). Optical steering angle sensor consists of an LED and a photodiode array or a CMOS line sensor. The sensor output is a digital square wave signal whose frequency depends on the speed of the turning wheel. Thus, the sensor can help determine the actual angle of rotation (ground truth value), which allows us to compare it with our proposed model. There are some mathematical tools that can adjust the steering angle values between different vehicles according to the formula of Equation (10):(10)$$R=\frac{s}{\sqrt{2-2\cos\left(\frac{2a}{n}\right)}}$$
where “R” is the radius of curvature, “*a*” is the steering wheel angle, “*n*” is the steering ratio, and “*s*” is the steering wheel frame. The onboard computer is NVIDA Jetson TX2. In the training phase, the captured images from ZED stereo cameras associated with steering angle are sent to this main computer and saved frame by frame as a custom image object. Later, the training can be accomplished offline using a more powerful computer than the onboard computer.

According to the steering angle sensor, we have many zero steering values and only a few left and right steering values. In fact, we mostly go straight, if we put all of them equal, then it will go left and right frequently. We cutoff 4402 images with zero values, which help us to balance the dataset, and the remaining 1224 stereo images are used to train the model as shown in Figure 4.

Many obstacles can be found in this dataset, such as bricks, dirt, newspapers, animals, and pedestrians, and even, we can see the nature of the road to provide driving assistance and more information. Naturally, there are many obstacles on the road, while we have deliberately placed a few obstacles to further complicate our dataset. There are total of 52 obstacles found in this dataset with variable height (8–45 cm). In addition, we also create the ground truth by manually driving the electric vehicle on the roadways. In order to obtain satisfactory driving model performance, it is necessary to train the model on multiple training dataset. Increasing the size of existing data through affine transformation [76], specifically by random change in contrast or brightness, horizontal flipping of the image, and random rotation. After development of autonomous vehicle driving model, we use the augmented dataset for training. The pseudocode for augmented dataset is included in Appendix A. This dataset is divided into two parts, 80% for training and 20% for validation, as shown in Table 2.

## 5. Results and Comparisons

We can use the following quantitative indicators to analyze the performance of the model: root mean square error (RMSE) and mean absolute error (MAE), as well as qualitative indicators through visualization: appearance challenges, distance and size challenges, clustering and shape challenges, and prediction of steering wheel angle challenges.

### 5.1. Quantitative Results

#### 5.1.1. Model Performance

Through our method and the measurement of RMSE and MAE, quantitative comparisons can be made to successfully identifying the obstacles. By comparing the RMSE and MAE values, we can predict the angle of the steering wheel. If the number of pixels of the obstacle is greater than the 50 pixels marked by the MRF model, the obstacle can be identified. If the pixel size of the segmented obstacle is higher than 50 pixels, it will be marked as a false alarm or high threat value and label it on the autonomous vehicle lane.

After successfully detecting the obstacles on the roadway by using the MRF model, we extracted the following two errors, root mean square error (RMSE) and mean absolute error (MAE), by which we can compare ground truth value with predicting steering wheel angle. For extracting these errors, we used following equations:(11)$$RMSE=\;\sqrt{\frac{1}{N}\overset{N}{\underset{i=1}{\sum }}{\left(G_{i}-P_{i}\right)}^{2}}$$
(12)$$MAE=\frac{1}{N}\overset{N}{\underset{i=1}{\sum }}{\left(\left|G_{i}-P_{i}\right|\right)}^{2}$$

Total number of images in the dataset is represented by “*N*”, ground truth represented by “*G_i_*”, and the steering wheel angle represented by “*P_i_*” for the image *i^th^* of the dataset.

We find out the RMSE and MAE values in different weather conditions and different times of the day and compare it with the ground truth value of the electric vehicle. In Figure 5, RMSE and MAE values are low in the following situation: daytime, shadow, and street light, while the error values are high in the following situations: night and rainy weather. The RMSE and MAE values are less than the value indicating that the predicted steering wheel angle closely follows the ground truth values.

#### 5.1.2. Path Follow

The Frenet Coordinate System [77] is used to analyze the situation avoiding the obstacles on the roadway of autonomous vehicle. We did not follow the Frenet Coordinate System for performance evaluation, but compared the ground truth value and RMSE value with time steps from 72,000 to 82,000 ms (represented on *x*-axis) and latitude (represented on *y*-axis). It can help us to understand that how autonomous vehicle closely follow the ground truth trajectory data and avoid the obstacle on the road as shown in Figure 6. As we can see in Figure 6, in the following situation: daytime, shadow, and under street light, the autonomous driving model closely follows the ground truth values while difference between the night and rainy weather is high. When camera performance declines at night or in rainy weather, it is a challenge to identify obstacles in such situations. Appendix A mentions code related to extracting the ground truth and predicting RMSE values of steering wheel for comparison.

### 5.2. Qualitative Results

Here, we will examine methods implemented for proposed dataset. We categorize them based on challenges related to appearance, distance and size, cluttering and shape, and steering wheel angle prediction.

#### 5.2.1. Appearance Challenges

The appearance is challenging to see in two ways. First, the deceptive texture by dirt or mud on the road due to rain or flat object (such as newspaper and cardboard box) is very different from other objects (such as rock and brick) that may mislead autonomous vehicle. Second, the appearance of obstacles seems to be its background. It is very challenging to detect obstacle due to week intensity gradient. However, the method we propose can solve the above problems and robustly extract obstacles. As you can see in Figure 7d, we can accurately identify obstacle even if the road is wet. Likewise, it can avoid the objects such as mud and shadow in Figure 7c. It also helps to identify obstacles that look like road (obstacle 4 in Figure 7d). Our method can also take into account local changes in the road structure and even applies to cement floors (Figure 7a,c).

#### 5.2.2. Distance and Size Challenges

As we all know, stereo depth is unreliable after the range of 10–12 m, which makes obstacle detection challenging at long distance. Detecting the small obstacles from distance, large changes in noise, or light reflection can cause serious problem. Our model solves these two problems well. As shown in Figure 7e, label 1 shows that our method has detected a 9 cm height obstacle at a distance of 8 m. Similarly, in Figure 7a, an obstacle of 7 cm height is detected at a distance of 7 m.

#### 5.2.3. Cluttering and Shape Challenges

Obstacle appear on the road in random shapes. Therefore, it makes sense to only allow certain shapes to use for modeled. However, our proposed algorithm has advantages in this regard because it does not require any assumptions about the shape or appearance of obstacles and always performs reliable detection. We can be seen rocks and tires in Figure 7b,c, respectively.

#### 5.2.4. Prediction of Steering Wheel Angle

The rule-based classification method is used to divide all obstacles into two categories: (a) the ones to be ignored (the pixel size of obstacle is less than 50) and (b) which causes of vehicle to collide (the pixel size of the obstacle is greater than 50). We use MRF model and segmented image to calculate the threat value using equation 9 as discuss in methodology section. After pixel segmentation of obstacles, the coordinates of the risky obstacles in the image are obtained.

After collecting the data, we prepare the image dataset for end-to-end driving model training by normalizing and resizing of the images [78,79]. As shown in Figure 8, the steering wheel angle is normalized between the −1.0 to +1.0, where positive values represent right rotation and a negative value represents the left rotation of the steering wheel. We normalize the values by using the liner transformation Equation (13):(13)$$\theta_{normalized}=\;-1.0+\max\left(0,\min\left(1.0,\;\frac{\theta_{raw}-\theta_{min}}{\theta_{max}-\theta_{min}}\right)\right)$$

Among them, $\theta_{normalized}$ is the normalized steering wheel angle between −1.0 and +1.0, $\theta_{raw}$ is steering wheel angle in radians, and $\theta_{min}\;\mathrm{and}\;\theta_{max}$ are the minimum and maximum steering wheel angle, respectively.

In the Figure 8, we have plotted the “steering angle” and “image sequence” on the y-axis and x-axis, respectively. Figure 8 shows that how closely proposed model follows the ground truth values. Using our model, we conducted a statistical significance test between the ground truth values and prediction values; we obtained a 91.5% confidence interval.

The accuracy of the prediction can be qualitatively represented by observing the direction of the driving angle of an autonomous vehicle as shown in Figure 9. As shown in Figure 9, when there are obstacles on the road, our proposed model can predict that the angle of the steering wheel will be closer to the ground truth angle. In Figure 9, the green line represents the ground truth angle and red line represents the expected angle of the steering wheel of our proposed model.

## 6. Conclusions

This research addresses how the robustness of obstacle detection method in a complex environment, by integrating a Markov random field (MRF) for obstacle detection, which is left intentionally or unintentionally on roadway, road segmentation, and CNN-model to navigate the autonomous vehicle safely. Our designed model predicts the steering wheel angle after the high threat value found of obstacle on the roadway of autonomous vehicle for both manual and self-driving vehicle. We conclude on the basis of analysis that our approach improves the safety of autonomous vehicle or human driving vehicle in terms of risk or collision with obstacles on the road by 35% as compared to vision-based navigation system rather than those who do not have the capability of detecting the high threat value obstacle on the road from long distances.

## Figures and Tables

**Figure 1 sensors-20-04719-f001:**
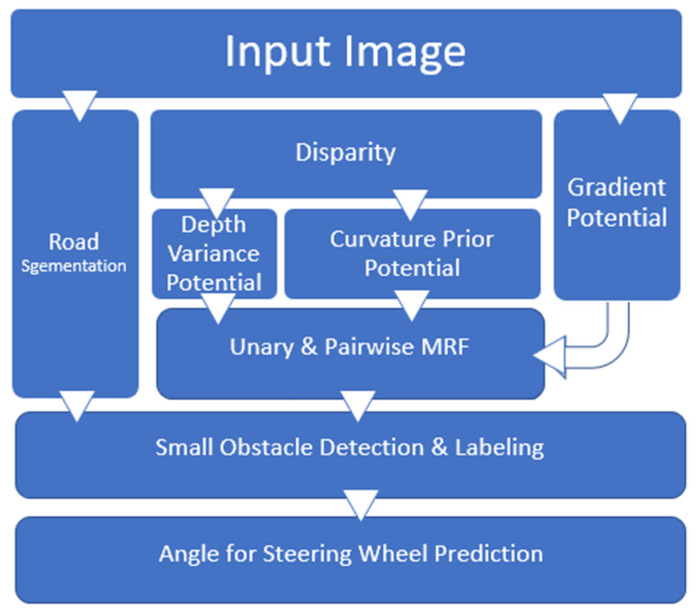
Pipeline of detection of hazard object and prediction of steering wheel angles.

**Figure 2 sensors-20-04719-f002:**
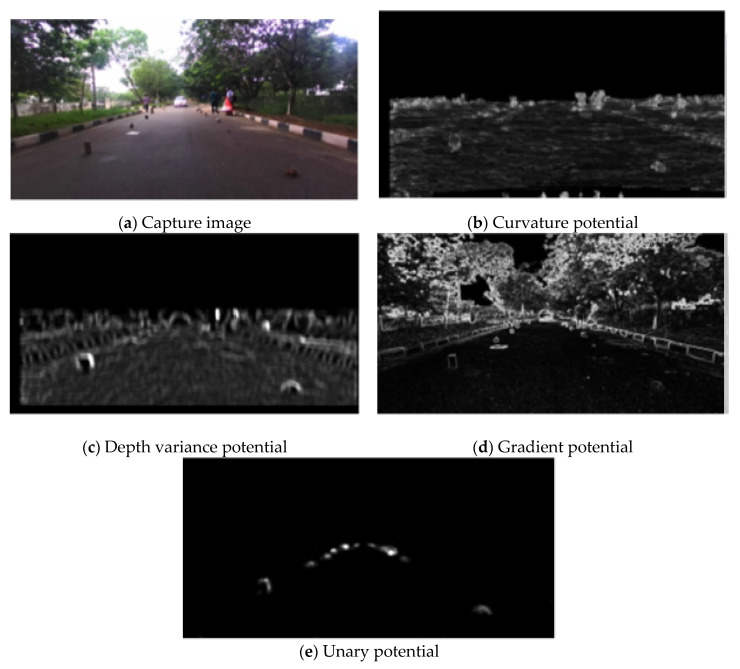
Result extracted from capture image by applying following potentials.

**Figure 3 sensors-20-04719-f003:**
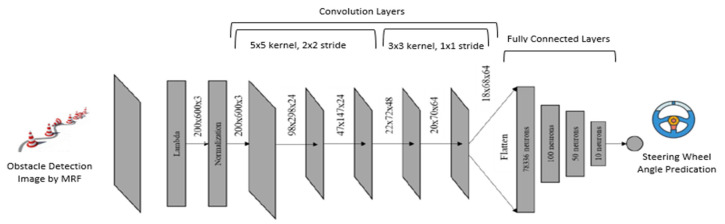
CNN autonomous vehicle driving model used for steering wheel angle prediction.

**Figure 4 sensors-20-04719-f004:**
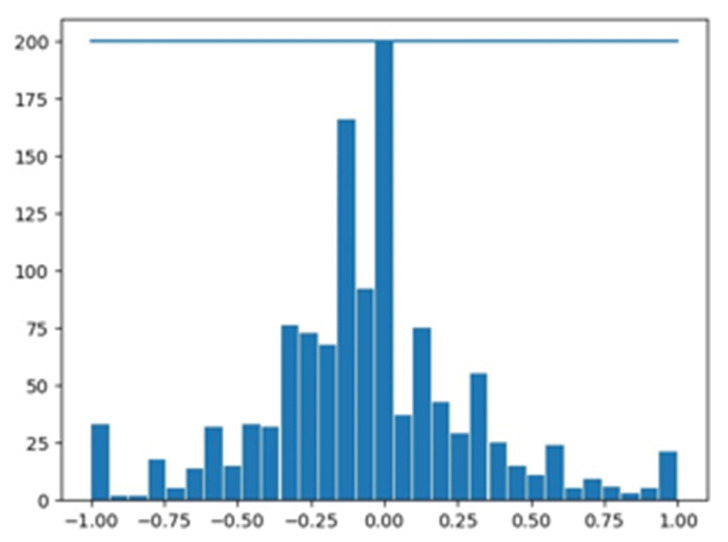
Balancing of the dataset shown in histogram graph.

**Figure 5 sensors-20-04719-f005:**
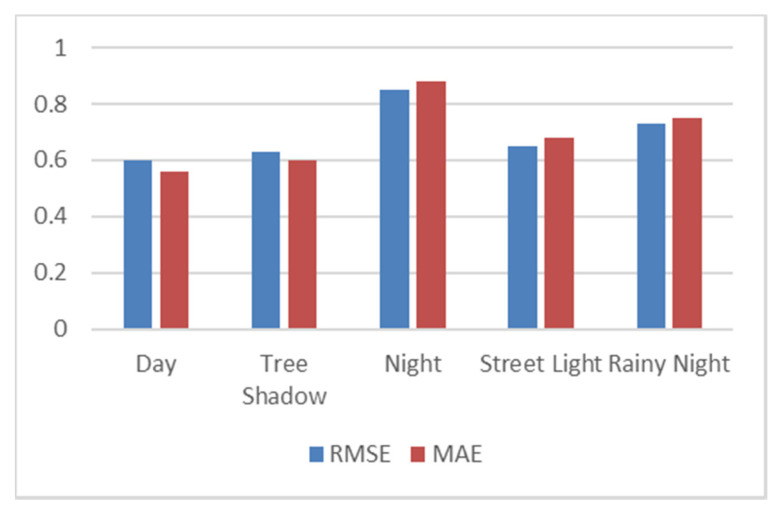
Measurement of root mean square error (RMSE) and mean absolute error (MAE) in different situation.

**Figure 6 sensors-20-04719-f006:**
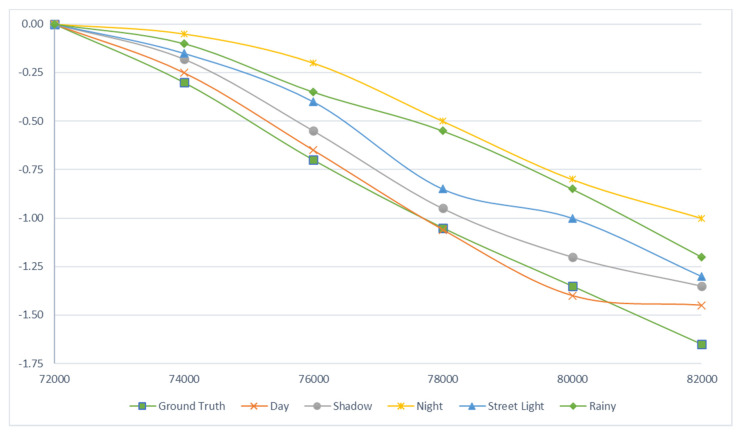
Comparison of the trajectory of the ground truth value with the RMSE value in different situation.

**Figure 7 sensors-20-04719-f007:**
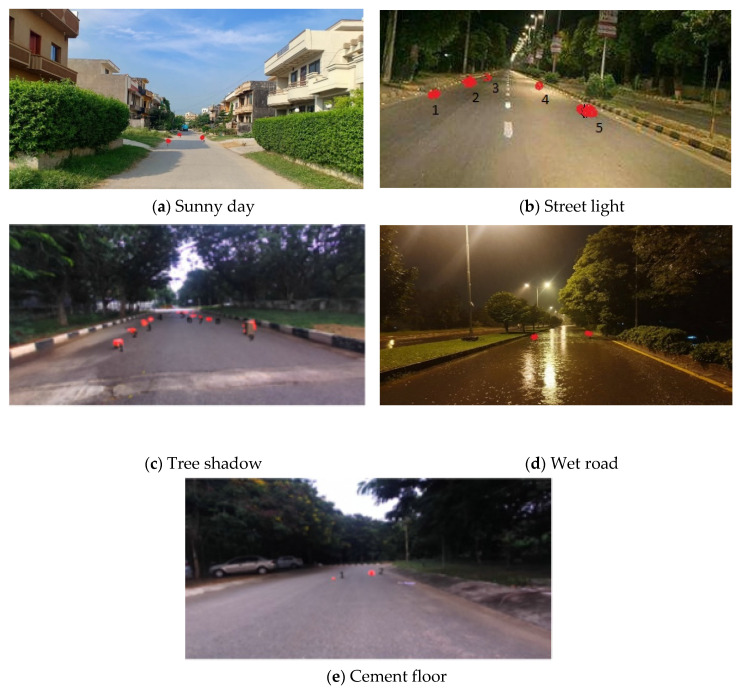
Obstacles in the captured image successfully identified by Markov random field (MRF) model.

**Figure 8 sensors-20-04719-f008:**
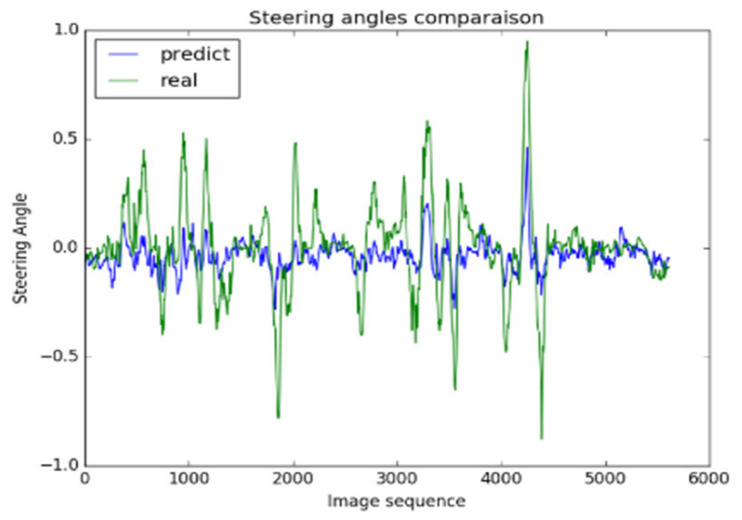
Comparison of ground truth steering wheel angle (real) with predicting steering wheel angle by model (predict).

**Figure 9 sensors-20-04719-f009:**
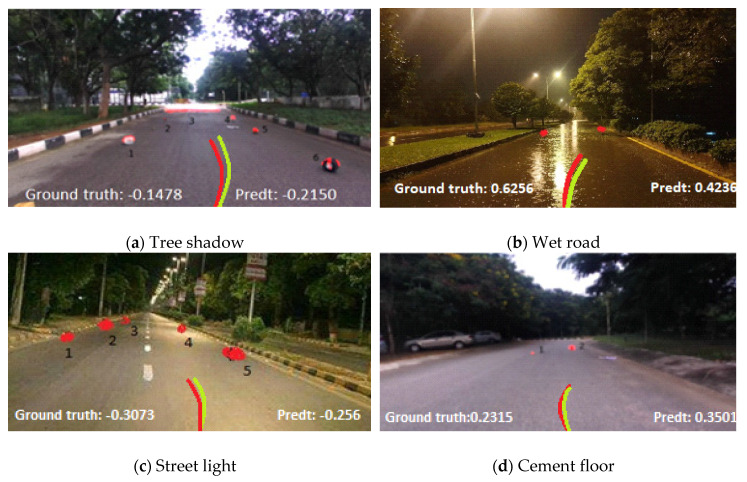
Comparison of ground truth angle with proposed model angle with random dataset images in different environment.

**Table 1 sensors-20-04719-t001:** Spec/parameters of ZED stereo device.

Camera	ZED M
Frame per second	30 fps
Resolution	2 × (1280 × 720)
RGB sensor type	1/3″ 4MP CMOS
Field of view	Max. 90° (H) × 60° (V) × 100° (D)
Exposure time	Set exposure to 50% of camera framerate
Focal length	2.8 mm (0.11″)—f/2.0
Interface	USB 3.0 Type-C port

**Table 2 sensors-20-04719-t002:** Dataset distribution for training and validation.

	Proposed Method	Dataset Size after Expansion
Size of dataset	1224	2448
Size of training dataset	980	1958
Size of validation dataset	244	490
Size of testing dataset (contain obstacle)	52	104

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
