# Peer review of "Obstacle Detection and Safely Navigate the Autonomous Vehicle from Unexpected Obstacles on the Driving Lane"

_sensors, 2020, doi:10.3390/s20174719_

Round 1

Reviewer 1 Report

Dr. Hou Jin and Malik Haris presented a description of their vision-based research approach to implement obstacle segmentation, as well as vehicle pose estimation, by leveraging Markov Random Field model. This methodology and the outcome by analyzing real world data accordingly may be of interest for both academia and self-driving car industry. However, the whole writing and the presentation appear as an in-progress, unfinished manner, e.g. there is no descriptive caption for figures and graphs.  

With that being said, a few major points should be addressed properly to improve the overall quality and readability of this work, before we consider it for publication. 

  1. (Line 85) "the fusion of DEMs based on stereo LiDAR have been studied for obstacles and edge detection tasks."  Stereo LiDAR is not a proper name. It can mislead readers. Does the author imply the fusion of stereo camera and LiDAR. More technical details, such as early fusion or late fusion, and reference papers, should be added.
  2. (Line 220-221) "This dataset contains 100 stereo images ... by using the ZED stereo device mounted on a Husky robot." More experimental details must be added, probably in a separate section called "Experimental Methods", such as all the specs/parameters about this ZED stereo device, camera exposure time, frame rate, etc. 
  3. Please add good and descriptive captions to all the figures and graphs, especially Figure 3, 4, 5, and Graph 1.
  4. (line 285-294). More details about the steering wheel angle estimation must be added. What is the meaning of the numbers in Figure 5 (ground truth, predt). How do you go these numbers? Online or offline? What is the frame rate of the camera? How long does it take you or the robot to process the data? During this period of time, does the robot move again? A comprehensive discussion about the data in detail should be provided.
  5. The authors should provided their software code or pseudo code in the appendix.   

Author Response

Question 1: 

(Line 85) "the fusion of DEMs based on stereo LiDAR have been studied for obstacles and edge detection tasks."  Stereo LiDAR is not a proper name. It can mislead readers. Does the author imply the fusion of stereo camera and LiDAR. More technical details, such as early fusion or late fusion, and reference papers, should be added.

Answer: I rewrote it and put all the related work in more details as you instructed. I also added some relevant reference papers.

---------------------------------------------------------------------------------------

Question 2: 

(Line 220-221) "This dataset contains 100 stereo images ... by using the ZED stereo device mounted on a Husky robot." More experimental details must be added, probably in a separate section called "Experimental Methods", such as all the specs/parameters about this ZED stereo device, camera exposure time, frame rate, etc. 

Answer: Currently I mounted ZED stereo camera on electric vehicle to get better results. I added the specifications and parameters of the ZED device in dataset section, as well as more relevant details about it. In method section, I described how we will use this data.

---------------------------------------------------------------------------------------

Question 3: 

Please add good and descriptive captions to all the figures and graphs, especially Figure 3, 4, 5, and Graph 1.

Answer: I have described all the figures, tables and graphs with more descriptive captions.

---------------------------------------------------------------------------------------

Question 4: 

(line 285-294). More details about the steering wheel angle estimation must be added. What is the meaning of the numbers in Figure 5 (ground truth, predt). How do you go these numbers? Online or offline? What is the frame rate of the camera? How long does it take you or the robot to process the data? During this period of time, does the robot move again? A comprehensive discussion about the data in detail should be provided.

Answer:

  • I added detailed information about generating steering wheel angle in the dataset section.
  • In Figure 5, the ground truth and prediction show the normalized steering wheel angle between -1.0 to +1.0 for comparison of manually generated steering wheel angle with the prediction steering wheel angle. I’ve added more information in 2.4 section, how to normalize this value between -1.0 to +1.0.
  • The training can be accomplished offline using a more powerful computer than the onboard computer.
  • Frame rate of the camera is 30fps.
  • I rewrite the dataset more in details

---------------------------------------------------------------------------------------

Question 5: 

The authors should provided their software code or pseudo code in the appendix.  

Answer: I added the pseudo code in method section “Algorithm 1” and also provided software code in appendix.

Reviewer 2 Report

This paper proposed an obstacle detection and safely navigate method for autonomous vehicles. The method discuss MRF model by fusing three potentials. The paper is practical and rational in some degree. However, there is still some major changes should be considered.

  1. In the Introduction section, the structure is strange. The authors should introduce the background, analysis related research literatures, present the motivations and contributions of proposed method. However, the authors do not give a good introduction, the authors give too much useless description of CNN in paragraph 3 and even do not mention the contributions. Thus, the authors should make a revision and rewrite the introduction section.
  2. In the related work section, the authors do not give enough papers to review the related research work. Indeed, some of the literatures are not very suitable for this paper like the description about detecting small obstacles in indoor scenes, which is very different with the autonomous driving situations.
  3. Figure 1 is too simple to serve as the main pipeline of this paper. The authors should redraw a more detailed and representative of the proposed detection and prediction pipeline.
  4. In Results and Comparisons section, the authors should add more quantitative results and compare the proposed method with other competitive algorithms.
  5. The number of references paper is 29 which is not enough. I recommend the authors add some new references.
  6. There are many typos and grammar errors, and the authors should find an English native speaker to help revise the manuscript.

Author Response

Question 1:

In the Introduction section, the structure is strange. The authors should introduce the background, analysis related research literatures, present the motivations and contributions of proposed method. However, the authors do not give a good introduction, the authors give too much useless description of CNN in paragraph 3 and even do not mention the contributions. Thus, the authors should make a revision and rewrite the introduction section.

Answer: I rewrote it and include all the relevant content as you mentioned (report analysis, current technology used, challenges and proposed method). I rewrote the CNN model paragraph and mentioned the contributions in our model.

-----------------------------------------------------------------------------------------

Question 2:

In the related work section, the authors do not give enough papers to review the related research work. Indeed, some of the literatures are not very suitable for this paper like the description about detecting small obstacles in indoor scenes, which is very different with the autonomous driving situations.

Answer: I rewrote it and put all the related research work in it as you instructed. I added some more relevant papers which are suitable for my work.

-----------------------------------------------------------------------------------------

Question 3:

Figure 1 is too simple to serve as the main pipeline of this paper. The authors should redraw a more detailed and representative of the proposed detection and prediction pipeline.

Answer: I redraw Figure 1 in more detail. I hope it is convenient for readers to understand the pipeline of this paper.

-----------------------------------------------------------------------------------------

Question 4:

In Results and Comparisons section, the authors should add more quantitative results and compare the proposed method with other competitive algorithms.

Answer: I followed your instruction and added more quantitative results and compared the proposed method in different situations for the autonomous vehicle.

-----------------------------------------------------------------------------------------

Question 5:

The number of references paper is 29 which is not enough. I recommend the authors add some new references.

Answer: I added 53 more reference paper in my work. Now, total count of reference paper are 82.

----------------------------------------------------------------------------------------

Question 6:

There are many typos and grammar errors, and the authors should find an English native speaker to help revise the manuscript.

Answer: I have tried my best to keep my English correct throughout this paper. I also showed my paper to English native speaker to help correct the grammatical errors.

Round 2

Reviewer 1 Report

Thanks to authors' prompt response and good improvement.

Reviewer 2 Report

The revision of this manuscript answers all my question, which satisfy the requirement of a regular paper. I recommend to accept this manuscript.